# Using an achiasmic human visual system to quantify the relationship between the fMRI BOLD signal and neural response

**Pinglei Bao[1], Christopher J Purington[2,3,4], Bosco S Tjan[1,4]***

[1]Neuroscience Graduate Program, University of Southern California, Los Angeles, United States; [2]School of Optometry, University of California, Berkeley, Berkeley, CA, United States; [3]Vision Science Graduate Program, University of California, Berkeley, Berkeley, United States; [4]Department of Psychology, University of Southern California, Los Angeles, CA, United States

**Abstract** Achiasma in humans causes gross mis-wiring of the retinal-fugal projection, resulting in overlapped cortical representations of left and right visual hemifields. We show that in areas V1-V3 this overlap is due to two co-located but non-interacting populations of neurons, each with a receptive field serving only one hemifield. Importantly, the two populations share the same local vascular control, resulting in a unique organization useful for quantifying the relationship between neural and fMRI BOLD responses without direct measurement of neural activity. Specifically, we can non-invasively double local neural responses by stimulating both neuronal populations with identical stimuli presented symmetrically across the vertical meridian to both visual hemifields, versus one population by stimulating in one hemifield. Measurements from a series of such doubling experiments show that the amplitude of BOLD response is proportional to approximately 0.5 power of the underlying neural response. Reanalyzing published data shows that this inferred relationship is general.

*For correspondence: btjan@usc.edu

**Competing interests:** The authors declare that no competing interests exist.

## Introduction

Functional magnetic resonance imaging (fMRI) based on the blood oxygenation level dependent (BOLD) signal has provided unprecedented insights into the workings of the human brain. The quantitative relationship between neural signals and the fMRI BOLD response is not precisely known and remains an active area of investigation. Most studies using the BOLD signal to infer brain activity rely on analytical methods (e.g., the general linear model) that assume a linear relationship between the BOLD signal and neural response, despite noticeable deviations from linearity (*Boynton et al., 1996*).

The BOLD signal is indirectly related to local neural response through mechanisms associated with oxygen metabolism and blood flow (*Davis et al., 1998*; *Hoge et al., 1999*; *Thompson et al., 2003*; *Griffeth and Buxton, 2011*). The neural response that is associated with information processing is itself multi-faceted. It comprises several interacting components, including subthreshold and suprathreshold electrical activities, the transport, release and reuptake of neurotransmitters, and various maintenance activities. Each of these components has its own metabolic and hemodynamic consequences. The common extracellular measurements of neural response include single- and multi-unit spiking activities and local field potential (LFP). While seminal studies have demonstrated a close relationship between the BOLD signal and these extracellular measurements of neural response (*Logothetis et al., 2001*; *Mukamel et al., 2005*), the quantitative nature of this relationship has not been sufficiently characterized. More importantly, since the relationship between these extracellular

**eLife digest** When a part of the brain becomes active, more oxygen-rich blood flows to it to keep its neurons supplied with energy. This flow of blood can be measured using a technique called functional magnetic resonance imaging (fMRI). Yet, it was not known exactly how the magnitude of the signal recorded from the oxygenated blood flow – dubbed the BOLD (blood oxygenation level dependent) signal – relates to the level of neural activity.

In most people, the brain area that processes fundamental visual information – called the visual cortex – receives signals from both eyes, sent via the optic nerves. The two eyes' optic nerves are bridged together with a structure called the optic chiasm, which ensures that each side of the brain gets input from both eyes for one side of the visual field. However, in rare cases, a person may lack an optic chiasm, and instead each side of the brain processes information about both sides of the visual field seen by one eye. This condition is known as achiasma.

Bao et al. have now used fMRI and behavioral experiments to study the brain activity of a volunteer who lacks an optic chiasm. This revealed that each half of the visual field stimulates different neurons in the same brain hemisphere of an achiasmic visual cortex. The two sets of neurons do not interact with each other, but they do share the same local blood supply. Moreover, these sets of neurons are organized in such a way as to preserve normal vision, and can be controlled independently using visual stimulation.

If both sets of neurons are stimulated with the same visual input at the same time, they together trigger twice as much neural activity as when just one set is stimulated. This also causes an increased BOLD signal as more blood flows to that region of the brain. Bao et al. were therefore able to infer a mathematical relationship between neural activity and the BOLD signal. This revealed that the magnitude of the BOLD signal is proportional to the square root of the underlying neural activity. Reanalyzing previously published BOLD data from other fMRI studies of healthy humans and monkeys supports this conclusion.

Bao et al.'s study provides scientists with a human model for noninvasively studying the origins and neural underpinnings of fMRI measurements, which may change how we analyze and interpret brain-imaging results in the future. The biggest challenge that researchers will likely face is in recruiting individuals with this rare condition of achiasma.

measurements and the intracellular components of neural activity is complex, the measured relationship between the BOLD signal to any specific extracellular components (e.g., power in the gamma band of LFP) may not reflect the relationship between the BOLD signal and the totality of neural response.

Most applications of fMRI, particularly in human neuroscience, sidestep any need for explicitly estimating neural activity and instead rely on establishing a direct relationship between the BOLD response and the stimulus condition. The general approach is to assume the BOLD responses evoked at different times and in different stimulus conditions sum linearly. Boynton and colleagues (1996) studied how the BOLD signal varied with the contrast and duration of stimulus presentation in the striate cortex and found that the system is approximately linear, in the sense that the BOLD response evoked by a 12 s stimulus was well approximated by summing the responses from two consecutive 6-s stimulations, even though predictions based on stimulations of much shorter durations (e.g., 3 s) failed to accurately predict the long-duration stimulus response. While this and similar studies (*Cohen, 1997*; *Dale and Buckner, 1997*; *Heckman et al., 2007*) have clearly noted the lack of linearity, their general message of an approximately linear system has nevertheless been used to justify the broad application of the general linear model (GLM) in fMRI data analyses. While the neural response is not explicitly involved in this type of analysis, it is always in the background — any nonlinearity observed in the BOLD response, e.g., in surround suppression or adaptation (*Grill-Spector and Malach, 2001*; *Kourtzi and Huberle, 2005*; *Larsson and Smith, 2012*) is often attributed to the underlying nonlinear neural response. The implicit assumption in common practice is that the relationship between the BOLD response and the neural response is essentially linear, a view that is widespread (*Logothetis and Wandell, 2004*) but under-examined.

An extensive set of biophysical models has been proposed to express either the steady-states (*Davis et al., 1998*; *Griffeth and Buxton, 2011*) or the dynamics of the BOLD response (*Buxton et al., 1998*; *Mandeville et al., 1999*; *Feng et al., 2001*; *Toronov et al., 2003*; *Blockley et al., 2009*; *Kim and Ress, 2016*) in terms of more basic physiological components, such as blood flow, blood volume, oxygen saturation, and oxygen extraction fraction in different vascular compartments. These biophysical models are foundational in our understanding of the BOLD signal, yet they do not provide any explicit and quantitative linkage between the neural response and the physiological components that are the inputs to these models. *Friston et al. (2000)* (see also *Stephan et al., 2007*), proposed a linkage between the evoked neural response and the blood-flow parameter of the Balloon model by *Buxton et al. (1998)*. While the resulting model is a powerful tool for inferring effective connectivity between brain regions from the BOLD signal, direct empirical support for this specific linkage is limited.

How could we empirically determine the quantitative relationship between the BOLD signal and the neural response, and do so when the constituents of the neural response are not comprehensively defined? A condition known as achiasma or non-decussating retinal-fugal fibre syndrome may provide an excellent model system for this purpose. This congenital condition prevents the normal crossing of optic nerve fibers from the nasal hemi-retina to the brain hemisphere contralateral to the eye (*Apkarian et al., 1994*; *1995*). The result is a full representation of the entire visual field (as opposed to only half the visual field) in each cerebral hemisphere (*Williams et al., 1994*; *Victor et al., 2000*; *Hoffmann et al., 2012*; *Davies-Thompson et al., 2013*; *Kaule et al., 2014*). Specifically, the representations of the two visual hemifields are superimposed in the low-level visual areas (V1-V3) ipsilateral to each eye, such that two points in the visual field located symmetrically across the vertical meridian are mapped to the same point on the cortex (*Hoffmann et al., 2012*). In other words, there are two pRFs for every point on this person's low-level visual cortex. The two pRFs are symmetrically located across the vertical meridian. Prior to the current study, it was not known if these pRFs were represented by one or two neural populations, or if these neural populations interacted.

In the current study, we found that the two pRFs are each represented by an independent population of neurons. The result is an in-vivo system with two independent populations of spatially intermingled neurons that share the same local control of blood vasculature. Because their population receptive fields (pRFs) do not overlap, an experimenter can independently stimulate each population by presenting a stimulus to its respective receptive field. Such a system is ideal for characterizing the relationship between neural and BOLD responses. Even though we may not know the constituents of the neural response, it will be reasonable to assume that the local neural response evoked by presenting identical stimuli to both pRFs, thereby activating both neuronal populations equally, is twice the neural response evoked by presenting the stimulus to just one of the pRFs. Measuring BOLD responses under these conditions allows us to not only directly test for linearity between the BOLD signal and neural response but also quantify the relationship between them, up to an arbitrary scaling factor. This approach does not require us to know the constituents of neural activity, and it is non-invasive.

To determine the relationship between neural response and the corresponding fMRI BOLD signal, we measured BOLD responses in the cortical areas V1-V3 of our achiasmic subject to luminance-defined stimuli. We presented stimuli of different contrasts to either one or both of the pRFs. From this data set, we used a model-free non-parametric method to infer the quantitative relationship between the BOLD signal (B) and neural response (Z). We found that the resulting B vs. Z function is well approximated by a power function with an exponent close to 0.5. The exponent stayed the same for short and long stimulus durations. We successfully cross-validated this result by comparing the inferred neural responses from this and twelve other fMRI studies to the single-unit responses obtained from non-human primates in similar contrast-response experiments.

## Results

### The achiasmic subject (S) and his retinotopy

Our primary research subject (S) was a 24-year-old achiasmic Caucasian male. S was diagnosed with isolated foveal hypoplasia. His uncorrected visual acuity was 0.7 logMAR for the left eye and 0.5

logMAR for the right eye (best-corrected 0.5 and 0.3 logMAR, respectively). He has a congenital nystagmus. His nystagmus (peak-to-peak) amplitude was less than 2.1° (95 percentile) horizontally and negligible (0.7°) vertically. The horizontal nystagmus followed a saw-tooth waveform with a right-forward fast phase and a (horizontal) frequency of between 0.6–2.3 Hz (mode at 2.3 Hz) during tasks performed for the current study. He was right-eye-preferred and suppressed his left eye (self-report). A Single Cover Test revealed left-eye esotropia. He could not achieve binocular fusion and had no measurable stereoacuity.

We confirmed his absence of an optic chiasm using MRI (*Figure 1A*). We obtained high-resolution monocular hemifield retinotopic maps for each of his eyes. The retinotopy was well defined and of high quality. Within the retinotopically defined visual areas V1-V3, the subject's retinotopic representation of the ipsilateral visual field is a mirror image of its contralateral field representation (*Figure 1B*). The two are superimposed on the hemisphere ipsilateral to the stimulated eye. Two points placed in the subject's visual field symmetrically across the vertical meridian are mapped to the same point on the cortex. This left-right mirror mapping is established at the level of individual voxels – 3x3x3 mm$^3$ (*Figure 1C*). Additional tests with vertically-reflected stimuli showed nearly identical BOLD signal modulation for individual voxels in V1-V3 (*Figure 1—figure supplement 1*). These findings were consistent with previous reports on human achiasma (*Hoffmann et al., 2012*).

The slight but systematic deviation from perfect homotopy in terms of eccentricity (*Figure 1C*, top row) is likely due to his asymmetric nystagmus waveform (with a leftward slow phase), which may have biased the time-averaged gaze position slightly towards the left of fixation. For those experiments that aimed to measure the interactions between the stimuli presented to both of the hemifields, we designed the experiments to be insensitive to this and other deviations from perfect gaze control or homotopy. Whenever appropriate, we made one of the stimuli (mask) much larger in size than the other (probe) to ensure that the probe was always projected to a cortical region that also represented the mask. When two stimuli had to be of the same size, we analyzed only the cortical regions associated nominally with the middle regions of the stimuli where overlaps were mostly guaranteed. We also conducted supplementary analyses to validate that the fMRI voxels used in the primary analysis were jointly activated by both stimuli.

## Co-locating but non-interacting neurons

Retinotopic mapping showed that each fMRI voxel in the visual cortex of S has two pRFs. We next tested if the neural populations that underlie these pRFs interact with each other. Behaviorally, S does not show any confusion between visual hemifields. Most tellingly, S is an avid reader, which strongly suggests that the co-localized neural populations do not interact functionally (otherwise, the text on the left of fixation would mask the text on the right of fixation). To quantify possible interactions between the pRFs, we measured whether the contrast threshold for S to detect a Gabor grating could be affected by placing a high-contrast flickering checkerboard symmetrically across either the vertical or horizontal meridian from the target location. Placing the checkerboard mask symmetrically across the vertical meridian causes it to be projected to the same cortical location as the target in the visual cortex of S, while placing it across the horizontal meridian does not (*Figure 2A*). S performed this sensitive threshold task monocularly with his right eye in front of a calibrated CRT screen. We found that the placement of the flickering checkerboard did not affect the detection threshold, and that the threshold is within the normal range without masking (*Figure 2B*). This and other experiments performed by us (*Figure 2—figure supplement 1*) and others (*Victor et al., 2000*; *Hoffmann and Dumoulin, 2015*) have failed to demonstrate any anomalous interactions between the two pRFs, strongly suggesting that the two underlying neural populations are functionally independent.

To test if the two neural populations are physiologically independent, we measured long-duration fMRI adaptation effects (*Fang et al., 2005*; *2007*). We first established that identical stimuli presented separately to either of the pRFs yielded nearly identical BOLD responses (*Figure 1—figure supplement 1*). We then measured the effect of adaptation when S was monocularly performing a highly demanding fixation task. The adapting and testing patterns were four counter-flickering Gabors (*Figure 2C*). Relative to the adaptors, the test patterns could be either at the same spatial locations or at the mirrored locations across the vertical meridian. The orientation of the test stimulus could be either the same as or orthogonal to the adaptors. As expected, when the adaptors and test patterns were at the same spatial locations, we observed strong BOLD adaptation when they

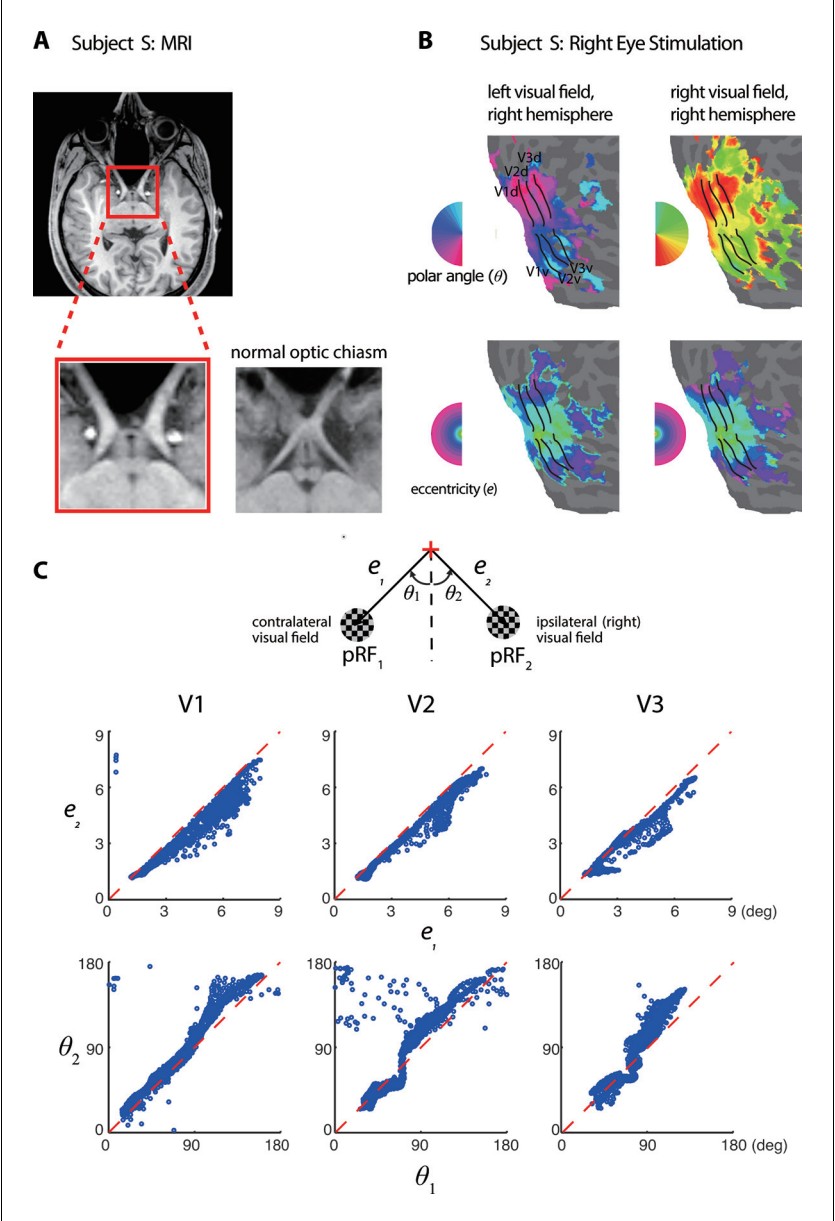

**Figure 1.** Retinotopy of the achiasmic subject S. (**A**) The MRI image of S shows the lack of the optic chiasm. (**B**) S has a well-defined retinotopy, but unlike normal retinotopic representations, both left and right visual fields are mapped to the same hemisphere ipsilateral to the stimulated eye (data from right-eye stimulation are shown). Eccentricity and polar angle maps are organized orderly for both visual hemifields. Visual area boundaries were identified at where the polar angle reversed. (**C**) A schematic of the two population receptive fields (pRFs) of a fMRI voxel. For individual voxels in V1-V3, eccentricities (upper panels) and polar angles relative to the lower vertical meridian (lower panels) of each of the two pRFs are plotted against each other. Most values fall close to the identity line, demonstrating that the two pRFs of each voxel are at mirrored locations across the vertical meridian.

The following figure supplement is available for figure 1:

**Figure supplement 1.** fMRI BOLD response evoked with ROI-defining stimuli.

were of matching orientation and a large release from adaptation when they were of orthogonal orientations. The critical conditions are when the adapting and testing patterns were at different spatial locations but projected to the same points on the cortex of S. In these conditions, regardless of orientation, we found a large release from adaptation equal or greater in amplitude than in the same-location cross-orientation condition. We take this result to mean that the co-localized neural

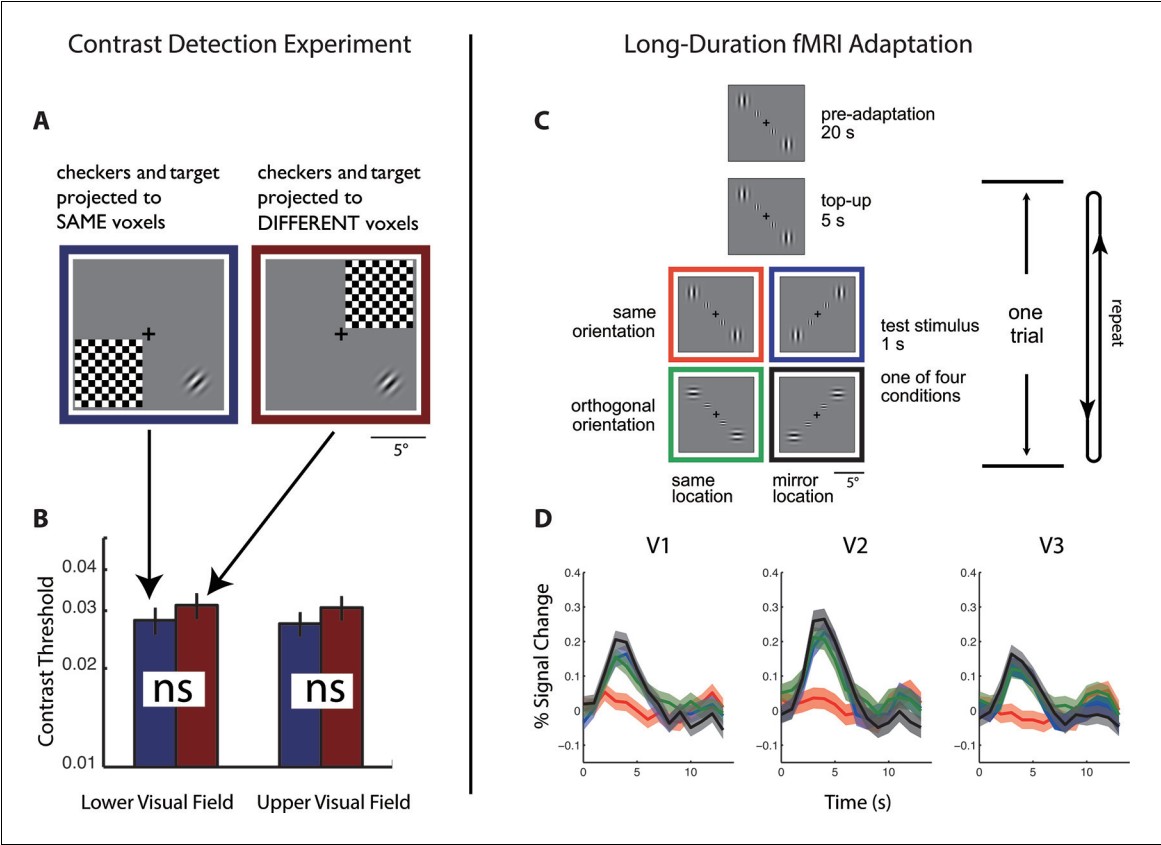

**Figure 2.** Behavioral and physiological evidence of independence between the co-localized neural populations. (**A**) Schematic of the psychophysical experiment of contrast detection with a contralateral mask. The 45° target Gabor patch was presented in either the lower-right (shown) or upper-left quadrant. A high-contrast flickering checkerboard mask was presented at the mirrored location from the target quadrant, across either the vertical or horizontal meridian. S was to identify which one of the two temporal intervals the target appeared in. (**B**) Contrast detection thresholds were essentially the same for the two mask-target arrangements. Error bars denote ± SE across blocks. (**C**) Design of the fMRI adaptation experiment. Each block of trials was preceded with 20 s of pre-adaptation. Each trial began with a 5 s presentation of the adapting stimulus ('top-up' adaptation), followed by one of the four test stimuli. Relative to the adapting stimulus, the test stimulus could either be at the same or mirrored location, and could have either the same or orthogonal orientation. Attention was controlled with a demanding central fixation task. (**D**) Time courses of fMRI BOLD responses in V1-V3 to the four test conditions. Shaded error band denote ± SE across trials. The responses evoked by the test stimuli presented at the mirrored location relative to the adaptor, regardless of orientation, did not differ significantly from those evoked by the test stimuli at the same location as the adaptor but with orthogonal orientation.

The following figure supplement is available for figure 2:

**Figure supplement 1.** Results from supplementary psychophysical experiments implicating two groups of non-interacting neurons.

populations that underlie the two pRFs on either side of the vertical meridian do not interact physiologically with each other.

## A 'pure' BOLD summation experiment

Our results thus far suggest that each fMRI voxel in the visual cortex of S contains two independent populations of neurons, each with a distinct pRF. To infer the relationship between neural response and the BOLD signal, we conducted a 'pure' BOLD summation experiment with S in which stimulus summation was absent. We stimulated each neural population separately with spatially disjoined stimuli A and B that were mirror images of each other (*Figure 3A*); we also stimulated them together by presenting A and B simultaneously (A+B).

Each stimulus consisted of four counter-flickering black-and-white checkerboards. The Weber contrast of the stimuli ranged from 0.05 to 1.0 in four equal log steps for a total of 5 contrast levels. In separate experiments, a stimulus was presented for either 1 s or 6 s, followed by a 16 s blank. We

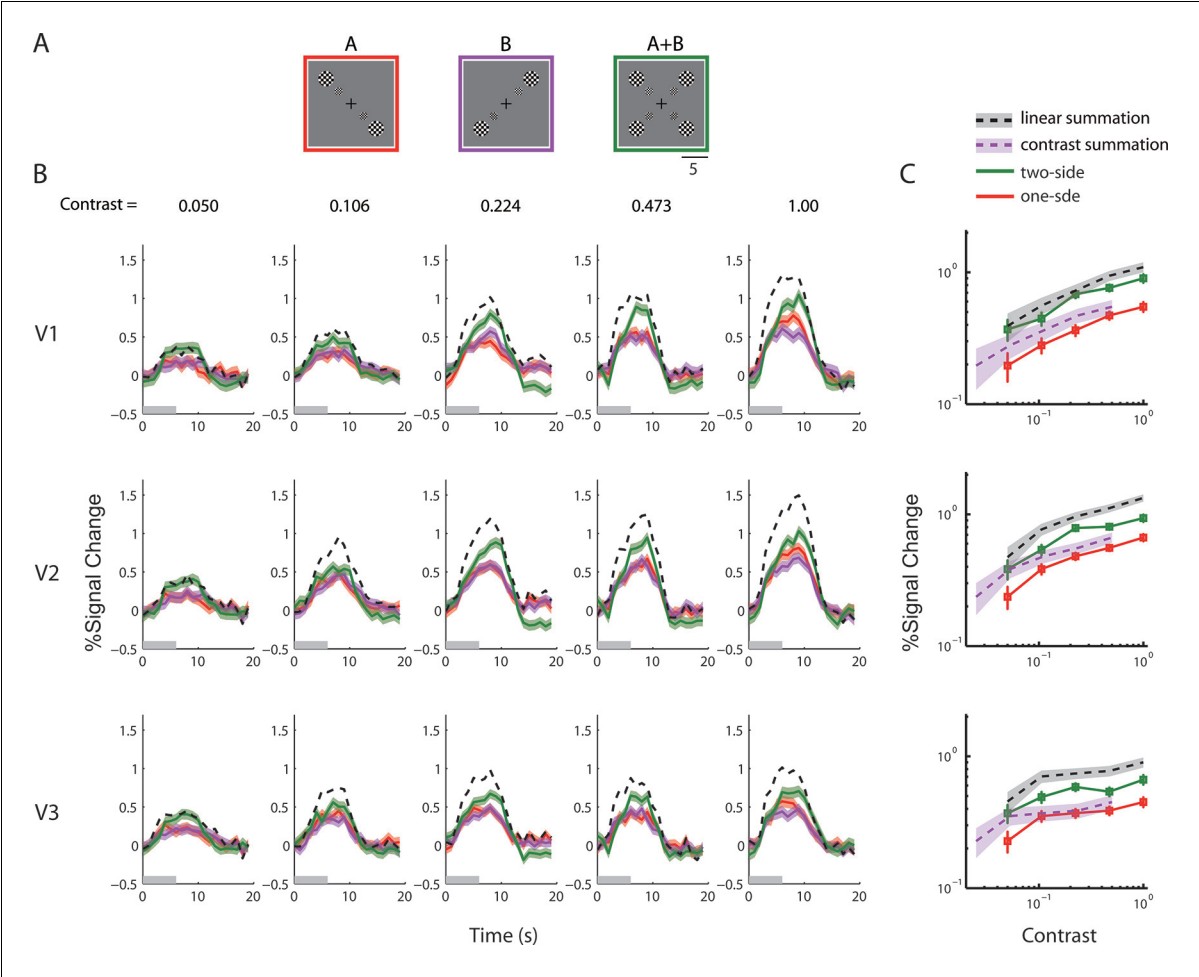

**Figure 3.** BOLD summation in the absence of neural nonlinearity associated with stimulus summation, with the 6-s stimuli. (See *Figure 3—figure supplement 1* for results obtained with the 1-s stimuli, which are qualitatively identical.) (**A**) The stimuli used in the BOLD summation experiment. The full stimulus display subtended 24°(w) x 19°(h). Stimulus types A and B are single-sided stimuli, while type A+B is a double-sided stimulus. BOLD responses associated with the outer checkerboard discs were extracted from the corresponding ROIs. These outer discs were of diameter 4° and centered at an eccentricity of 7°. (**B**) Estimated peristimulus time courses from V1-V3 for the three stimulus types at five different contrast levels. Red and magenta represent responses (lines) and ± SE (bands) to the single-sided stimuli, and green represents responses to the double-sided stimuli. Black dashed lines represent the predictions of linear BOLD summation, which overestimated the measured responses (green bands). Gray bars on the abscissa indicate the duration of the stimulus (6 s). (**C**) The contrast response functions of V1-V3 as defined by the amplitudes of the time courses. The amplitude of a time course was taken to be the average response between 7–9 s post-stimulus onset when the response typically reached its peak. The red lines represent the average single-sided response amplitudes as a function of luminance contrast. The green lines represent the average double-sided response amplitudes. The gray bands represent the predicted responses (68.2% confidence interval) evoked with the double-sided stimulus under the assumption of linear BOLD summation (i.e. the summed response to conditions A and B), and the magenta bands represent the prediction of contrast summation – the equivalent contrast of a double-sided stimulus being twice that of the corresponding single-sided stimulus. The measured double-sided responses were significantly lower than the predictions of linear BOLD summation and higher than that of contrast summation.

The following figure supplements are available for figure 3:

**Figure supplement 1.** fMRI BOLD summation with a 1-s stimulus duration.

**Figure supplement 2.** Results of the 6-s BOLD summation experiment with and without removing the global noise components.

**Figure supplement 3.** Results of the 1-s and 6-s BOLD summation experiments obtained from the corresponding retinotopically-defined V1 ROI in the left hemisphere of the achiasmic subject.

measured the full peristimulus time course of the evoked BOLD response and its amplitude. In V1-V3 and across the contrast levels, the BOLD response to the combined stimulus A+B was significantly higher than what could be produced by doubling the contrast of stimulus A or B [paired t-test; V1: $t(19)$ = -4.48, p=$2.57 \times 10^{-4}$; V2: $t(19)$ = -2.96, p=0.008; V3: $t(19)$ = -4.30, p=$3.85 \times 10^{-4}$] (*Figure 3C*) but significantly lower than the sum of the BOLD responses to stimuli A and B [paired t-test; V1: $t(24)$ = 2.53, p=0.018; V2: $t(24)$ = 6.20, p=$2.11 \times 10^{-6}$; V3: $t(24)$ = 6.07, p=$2.87 \times 10^{-6}$] (*Figure 3B*). This finding shows that BOLD summation, in the absence of nonlinearity associated with neuronal summation (assuming that the A+B stimulus simultaneously excited two independent populations of neurons), is itself nonlinear, with a compressive nonlinearity. The compressive nonlinearity was not due to response saturation since the 1-s stimulation duration yielded essentially the same nonlinearity as the 6-s simulation. This result implies a nonlinear relationship between neural and BOLD responses, and challenges the prevailing linearity assumption.

## Inferring the quantitative relationship between BOLD amplitude and neural response

We defined the amplitude of a BOLD response to be the peak of the BOLD time course. We sought to express BOLD amplitude (*B*) as a function of neural response (*Z*), where *Z* refers to the local aggregate of neural response with unspecified constituents. At each cortical region of interest, the BOLD summation experiment provided five levels of *Z*, corresponding to the five levels of luminance contrast for the single-sided stimuli (A, B). (The single-sided stimuli A and B resulted in nearly identical time courses (*Figure 3B*, see also *Figure 3—figure supplements 2* and *Figure 4—figure supplement 2* on robustness), indicating that the inputs from the left and right visual fields were balanced. We therefore use their averages in the analysis.) For each level of *Z* evoked by a single-sided stimulus, the level of neural response evoked by the corresponding double-sided stimulus (A+B) is doubled under the assumptions of co-localization and independence, which we have empirically validated. We therefore have five pairs of BOLD measurements that correspond to $Z_i$ and $2Z_i$ (*Figure 4A*, left column). We do not know the values of $Z_i$, but we can reasonably assume that these levels of neural response were ordered by stimulus contrast: $Z_1 \leq \ldots \leq Z_5$.

We proceed to 'stitch' the five pairs of measurement to form a continuous function without assuming any specific functional form, except that the resulting function should be monotonic and smooth. Without loss of generality, we can set $Z_1$ to 1. We are then left with four unknowns $1 \leq Z_2 \leq \ldots \leq Z_5$. We estimated the four ordered values $Z_2 \leq \ldots \leq Z_5$ such that the ten data point $(Z_1. B_{1,1}),\ldots,(Z_5. B_{1,5})$, $(2Z_1. B_{2,1}),\ldots,(2Z_5. B_{2,5})$ can be best described, in the least-squares sense, by a monotonic and smooth function (see Methods). This procedure amounts to shifting horizontally on a log-scaled abscissa a pair of data points $\{(Z_i. B_{1,i}), (2Z_i. B_{2,i})\}$ relative to other pairs until all five pairs (ten data points) fall on a smooth monotonic curve.

Applying this stitching procedure to BOLD amplitude data resulted in three BvZ functions (*Figure 4B*, left column) for V1, V2, and V3 respectively. While the stitching procedure did not *a priori* assume a specific functional form, it is clear from *Figure 4B* that the resulting BvZ functions can be well fitted by a power-law function (*Figure 4B*): $B = kZ^\gamma$, where *k* is an arbitrary scaling factor related to the unit of *Z* ($R^2$ = 0.997, 0.992, and 0.988 for V1, V2, and V3, respectively). The indeterminacy of *k* reflects the fact that we do not know the constituents of neural response. For our purpose, the critical parameter is the exponent (γ). This power-law relationship between neural and fMRI BOLD responses is also evident from the parallel lines in *Figure 4A* – there was no significant interaction between sided-ness (single vs. double) and contrast levels. An alternative approach for estimating γ by first determining the applicability of a power-law function is described in *Figure 4—figure supplement 1*; it resulted in essentially the same set of values for γ.

We found that for a stimulus duration of 6 s, γ = 0.76 [95% CI: 0.66, 0.87], 0.54 [0.48, 0.62], and 0.55 [0.46, 0.67] for V1, V2, and V3, respectively. If hemodynamics are essentially the same in V1-V3, then the exponent γ will be the same across these areas. The exponents were indeed statistically indistinguishable between V2 and V3; however, the value for V1 was outside the 95% confidence intervals of the values for V2 and V3 and vice versa (*Figure 4C*, left column). We believe that the measured γ value obtained in V1 was inflated towards unity (linearity) because the neural populations associated with the two pRFs are not truly co-localized in V1 (see Discussion). However, the assumption of co-localization is needed for inferring the BvZ function from the summation experiment. Given the good agreement in the γ value between V2 and V3, the true value of γ in V1 is likely

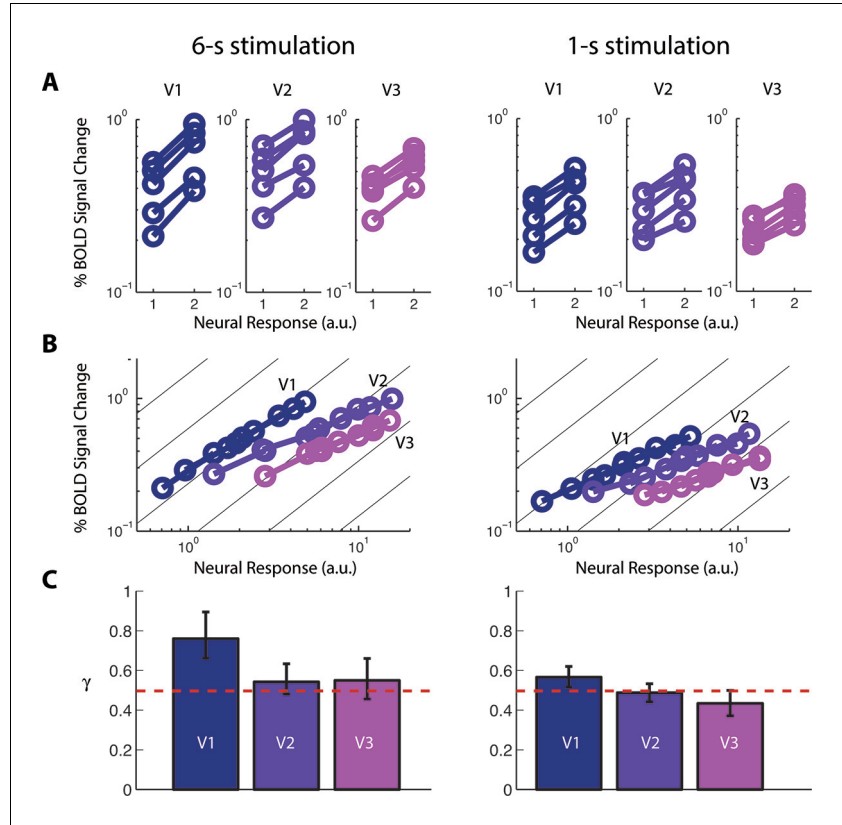

**Figure 4.** fMRI BOLD signal as a function of neural response. (**A**) Five pairs of BOLD response amplitudes evoked in V1-V3 with the single- and double-sided stimulations, each with two stimulus durations, 6-s (left column) and 1-s (right column). If the neural response to a single-sided stimulus is $Z_i$, then the neural response to the corresponding double-sided stimulus will be $2Z_i$, given our empirical determinations of co-localization and independence of the neuronal populations in an achiasmic visual cortex. (**B**) The BOLD vs. neural response (BvZ) functions for V1-V3 as inferred by the stitching procedure for the two stimulus durations. The inferred functions can be well fitted with power-law functions (i.e. straight lines in log-log coordinates). These functions are nonlinear, with a log-log slope significantly shallower than unity (the background gray lines). (**C**) The exponents ($\gamma$) of the power-law fit of the BvZ functions for V1-V3. Error bars denote 95% CI. The red line indicates $\gamma = 0.5$. $\gamma$ estimated from V2 and V3 ($\gamma \sim 0.5$) were not significantly different, while that obtained from V1 was biased upward, due to a violation of the co-localization assumption (see Discussion) required for inferring the BvZ function using the summation experiment. We thus inferred the (true) BvZ function of V1-V3 using the average $\gamma$ estimated from V2 and V3 only.

The following figure supplements are available for figure 4:

**Figure supplement 1.** Alternative derivation of the BvZ function.

**Figure supplement 2.** Robustness of BOLD summation results.

close to those estimated from V2 and V3. We thus combine the measurements from areas V2 and V3 to obtain the BvZ function with an exponent equal to $0.54 \pm 0.05$ for the 6-s stimulus. We believe this BvZ function is sufficiently general and applicable for at least the occipital lobe.

Changing stimulus duration changes the time course of the evoked neural response. If the BvZ function describes a general relationship between the neural response and fMRI BOLD signal, then the value of $\gamma$ should be relatively constant across stimulus durations. This is indeed the case. Reducing the stimulus duration from 6 s to 1 s resulted in only small changes to the exponent of the BvZ function, with $\gamma = 0.56$ [0.51,0.62], 0.49 [0.44,0.54], and 0.43 [0.37,0.50] for V1, V2, and V3, respectively (*Figure 4*, right column). As with the 6-s stimulus, $\gamma$ estimated from V1 is again significantly higher that those from V2 and V3, while those from V2 and V3 are not significantly different

from each other. Combining data from V2 and V3 yield $\gamma = 0.47 \pm 0.03$. Hence, a sixfold change in stimulus duration resulted in very little change in the nonlinearity. Combining data from the two stimulus durations, we have $\gamma \sim 0.5$.

## Comparing BOLD amplitude and spiking activity

Spike rate is one of the most common measures of neural response, and the BOLD response has been related to spike rate (*Heeger et al., 2000*; *Heeger and Ress, 2002*; *Logothetis and Wandell, 2004*). To cross-validate our finding and to make contact with the broader literature, we used the inferred BvZ function (with $\gamma$ inferred from V2 and V3) to estimate the neural response $Z$ from the BOLD amplitude data of the single-sided conditions in the BOLD summation experiment, which were typical contrast response measurements. The inferred neural activity in V1 for both the 6-s and 1-s stimuli matched extremely well with the average primate V1 contrast response function measured in terms of single-unit spiking activity by *Albrecht (1995)* (*Figure 5A*). Contrary to earlier reports based on the same single-unit data (*Heeger et al., 2000*), linearly scaling our BOLD amplitude data does not fit the single-unit spiking data. The nonlinearity in our data cannot be attributed to anticipatory and other endogenous responses that might be induced by the task structure (*Sirotin and Das, 2009*) (*Figure 3—figure supplement 3*). This is because our subject was engaged in a demanding central fixation task (orientation discrimination) that was asynchronous with the blocked contrast stimuli.

The BvZ function obtained from our achiasmic subject, with $\gamma \sim 0.5$, appears to represent the typical neurovascular coupling in humans. We arrived at this conclusion by considering 21 datasets from 12 published studies (*Table 1*) on the contrast response function measured with fMRI BOLD in human V1. We assume the BvZ function to follow a power-law relationship, as we have found with the achiasmic subject, but allowed the exponent ($\gamma$) to be a free parameter. $\gamma = 1$ represents a linear relationship between BOLD response and spike rate. We used two single-unit contrast response functions (spike rate vs. contrast) for this meta-analysis. One of the functions (*Figure 5B*) was from *Albrecht (1995)*, which was an average of 19 monkey V1 neurons, each tested with their respective optimal stimuli. The other (*Figure 5D*) was from *Heeger et al. (2000)*, which was averages of the predicted response (*Geisler and Albrecht, 1997*) of over 300 neurons to the specific stimulus used in *Boynton et al. (1999)* (Geisler, personal communication). For each BOLD data set, we estimated the value of $\gamma$ such that the resulting BvZ function, when applied to a single-unit contrast response function, resulted in a predicted BOLD response that best matched the BOLD data set in terms of an error-weighted least-squares fit. With respect to the two neuronal contrast response functions, the bulk of the distribution of $\gamma$ lies mid-range between 0 and 1 (*Figure 5B–E*). The overall picture given by this set of nearly two dozens experiments is that BOLD response is nonlinearly related to spike rate. The interquartile range of both distributions includes $\gamma \sim 0.5$, the exponent of the BvZ function independently estimated from our pure BOLD summation experiments with subject S. In fact, $\gamma \sim 0.5$ is near the median of one distribution (*Figure 5C*) and the mode of the other distribution (*Figure 5E*).

## Discussion

We found that the fMRI BOLD response amplitude is proportional to the local neural response raised to a power of about 0.5. We reached this conclusion by measuring, in the visual cortex of an achiasmic subject, fMRI BOLD amplitudes at five levels of neural activity *and* also at twice those levels. Our ability to double the local neural response relies on the presence of two co-localized but independent populations of neurons in the visual cortex of the achiasmic subject. The two neuronal populations are equally excitable, and each population has a distinct and non-overlapping population receptive field. We used fMRI retinotopy and localized stimulation to demonstrate co-localization and equal excitability. We used a sensitive contrast detection task and a long-duration fMRI adaptation task to demonstrate independence. Taken together, our results demonstrate that the achiasmic human visual cortex provides a versatile in vivo model for investigating the relationship between evoked neural response and the associated fMRI BOLD signal.

Reported cases of achiasma in the scientific literature are rare, particularly with regards to fMRI studies. *Victor et al. (2000)* performed a seminal study on the achiasmic visual system, the first one using fMRI and with a single subject. A more recent fMRI study, which provided a detailed

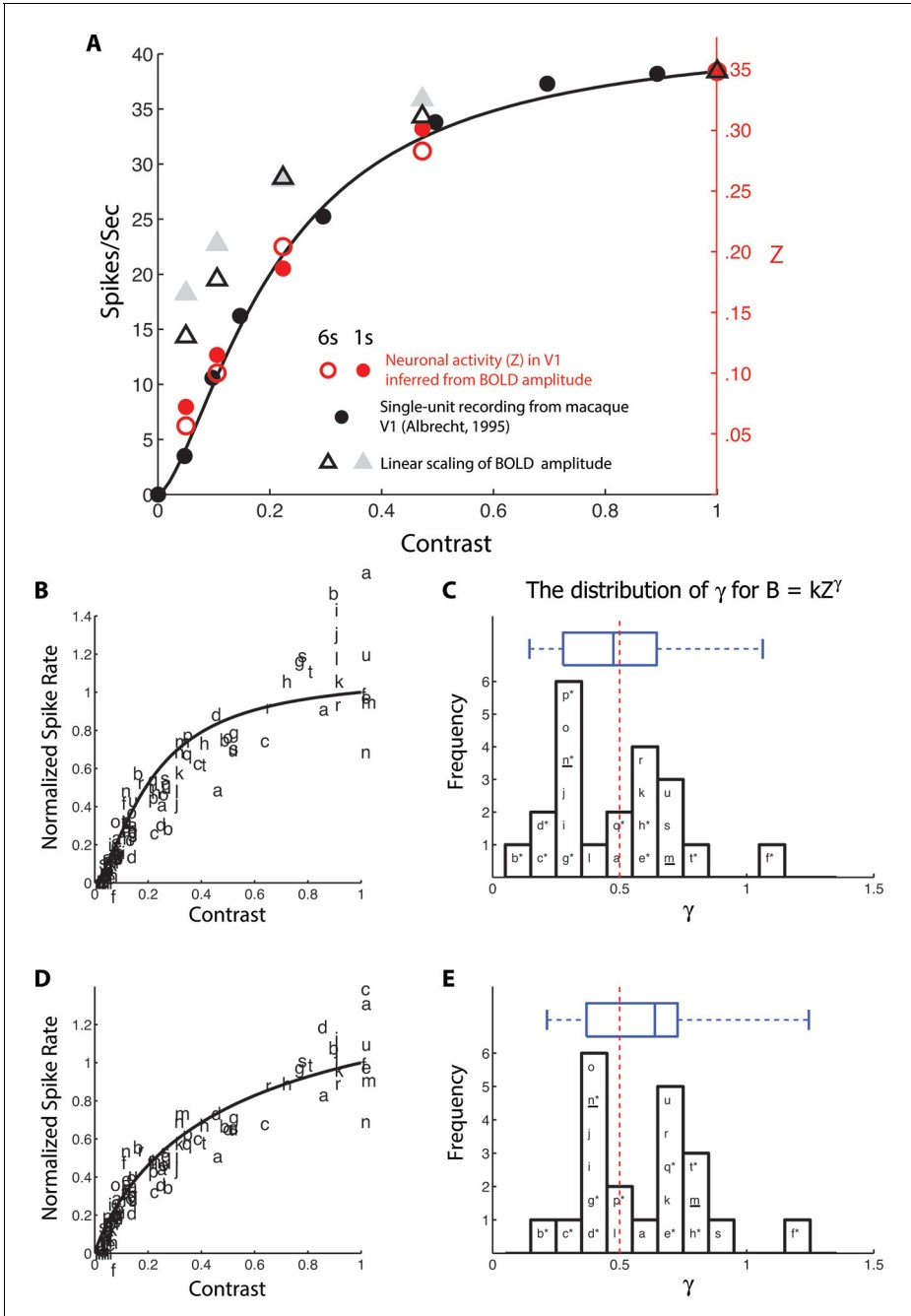

**Figure 5.** Comparisons between neural response inferred from the BvZ function ($B = kZ^\gamma$) and single-unit spiking activity. (**A**) Neural contrast response functions. The black dots and line are the average single-unit firing rate as a function of luminance contrast recorded in macaque V1 (replotted from *Albrecht, 1995*) and the best fitting Naka-Rushton function (*Naka and Rushton, 1966*), respectively. Red open and filled circles represent the BvZ-inferred neural contrast responses in V1 of the achiasmic subject, computed from our 6-s and 1-s single-sided BOLD data sets (from *Figure 3* and *Figure 3—figure supplement 1*) and matched to single-unit firing rates with a single scaling constant to align the data point at contrast = 1. Gray open and filled triangles represent the linearly scaled BOLD contrast responses from the same data sets. The BvZ-inferred neural responses are in excellent agreement with the single-unit spiking data, whereas linearly scaled BOLD responses are not. (**B**) fMRI BOLD contrast responses from 21 published data sets (a-u; see *Table 1*) were individually matched to the single-unit spiking responses function of (**A**), assuming a power-law function to convert BOLD response to spike rate. The fits were good, with R² ranging from 0.73 to 0.99. The distribution of the best-fitting exponents ($\gamma$) is shown in (**C**). The median of the distribution (0.48) is close to the exponent (0.5, red dashed vertical line) of the BvZ function inferred from the achiasmic subject. Asterisk (*) marks studies in which subjects attended to the stimuli used to obtain the contrast response functions. Underline (_) marks studies that measured and analytically discounted any task-related baseline response from the contrast response functions. (**D,E**) An otherwise identical analysis as in (**B,C**)

*Figure 5 continued on next page*

Figure 5 continued

but using the single-unit (spike rate) contrast response function obtained from *Heeger et al. (2000)* instead of the single-unit contrast responses function used in (**A**) from *Abrecht (1995)*. (Note that some data points are out of the ordinate range in **B** and **D**; they are omitted from the plots.)

characterization of the cortical representations of visual fields in achiasma, was conducted by *Hoffmann et al. (2012)* with two subjects, one in Germany and the other in the US. Another recent fMRI study on the achiasmic visual pathway was conducted by *Davies-Thompson et al. (2013)* with a single subject. None of these studies attempted to use achiasma as a model system to study the relationship between neural response and the BOLD signal, nor had they thoroughly assessed the neural underpinning of the overlapped visual-field representations in achiasma. Nevertheless, the fMRI visual field mapping results from these prior studies are very similar to our results, suggesting that our subject is a 'typical' achiasmic individual. Yet, among this rare patient population, our subject was unique. Aside from being otherwise neurologically normal, he was very dedicated to this research and made himself available for nearly a year to complete a large set of converging experiments, including replications.

**Table 1.** Data sources of *Figure 5*.

| Legend | Publication | Figure # (Condition) | Subjects | $\gamma$ (*Figure 5B*) | $\gamma$ (*Figure 5D*) | $R^2$ (*Figure 5B*) | $R^2$ (*Figure 5D*) |
|---|---|---|---|---|---|---|---|
| a | *Pestilli et al., 2011* | Figure 4B (Focal Cue – non Target) | Human | 0.50 | 0.64 | 0.77 | 0.86 |
| b* | *Pestilli et al., 2011* | Figure 4B (Focal Cue - Target) | Human | 0.15 | 0.21 | 0.93 | 0.94 |
| c* | *Pestilli et al., 2011* | Figure 4B (Distributed Cue –Target) | Human | 0.24 | 0.34 | 0.79 | 0.88 |
| d* | *Pestilli et al., 2011* | Figure 4B (Distributed Cue – non Target) | Human | 0.25 | 0.37 | 0.85 | 0.93 |
| e* | *Avidan et al., 2002* | Figure 2B (Faces) | Human | 0.61 | 0.72 | 0.96 | 0.93 |
| f* | *Avidan et al., 2002* | Figure 2B (Objects) | Human | 1.06 | 1.24 | 0.91 | 0.89 |
| g* | *Buracas & Boynton 2007* | Figure 1 | Human | 0.32 | 0.40 | 0.93 | 0.98 |
| h* | *Boynton et al., 1999* | Figure 3 and Figure 4 | Human | 0.64 | 0.84 | 0.97 | 0.99 |
| i | *Schumacher et al., 2011* | Figure 4A (gradient echo) | Human | 0.28 | 0.38 | 0.73 | 0.85 |
| j | *Schumacher et al., 2011* | Figure 4A (spin echo) | Human | 0.28 | 0.36 | 0.86 | 0.93 |
| k | *Schumacher et al., 2011* | Figure 4C (gradient echo) | Human | 0.56 | 0.66 | 0.97 | 0.98 |
| l | *Schumacher et al., 2011* | Figure 4C (spin echo) | Human | 0.40 | 0.50 | 0.91 | 0.96 |
| m | *Li et al., 2008* | Figure 3 (Unattended) | Human | 0.65 | 0.75 | 0.99 | 0.96 |
| n* | *Li et al., 2008* | Figure 3 (Attended) | Human | 0.26 | 0.36 | 0.91 | 0.85 |
| o | *Moradi & Heeger 2009* | Figure 2 | Human | 0.25 | 0.37 | 0.86 | 0.83 |
| p* | *Olman et al., 2004* | Figure 4A (Natural Images) | Human | 0.34 | 0.50 | 0.97 | 0.97 |
| q* | *Olman et al., 2004* | Figure 4A (Whitened Images) | Human | 0.48 | 0.68 | 0.99 | 0.97 |
| r | *Park et al., 2008* | Figure 9 | Human | 0.61 | 0.70 | 0.98 | 0.94 |
| s | *Tootell et al., 1998* | Figure 5 | Human | 0.71 | 0.87 | 0.89 | 0.94 |
| t* | *Zenger-Landolt & Heeger 2003* | Figure 9 (Without Surround) | Human | 0.75 | 0.83 | 0.85 | 0.93 |
| u | *Logothetis et al., 2001* | Figure 5B (BOLD) | Monkey | 0.65 | 0.68 | 0.75 | 0.88 |

Asterisk (*) marks indicate studies in which subjects attended to the stimuli used to obtain the contrast response functions.

Underline (_) marks indicate studies that measured and analytically discounted any task-related baseline response from the contrast response functions.

We are aware of the limitation of being a case study and have therefore included a meta-analysis of published data from neurologically normal participants. This meta-analysis was hypothesis-driven, based on results from our achaismic subject. The achiasma results informed us of what to look for – the exponent of the neural-BOLD function. Our meta-analysis found that the exponent is around 0.5, in agreement with our findings from the single achiasmic subject.

## Relationship between the fMRI BOLD signal and neural response

The quantitative relationship between the fMRI BOLD signal and neural response depends on the interacting hemodynamic quantities that are linked to neuronal activity: cerebral blood flow (CBF), cerebral blood volume (CBV) and cerebral metabolism rate of oxygen (CMRO2) (*Davis et al., 1998*; *Hoge et al., 1999*; *Logothetis et al., 2001*; *Thompson et al., 2003*; *Mukamel et al., 2005*). It also depends on the origin of the MR signal (e.g., intra- vs. inter-vascular space, capillaries vs. pial vessels), which in turn depends on the pulse sequence and magnetic field strength used to make the measurement (*Griffeth and Buxton, 2011*). The BOLD signal is inherently measurement-dependent, and there is not a 'canonical' BOLD signal. For the common applications of fMRI in human neuroscience, it is arguably more important to understand and quantify at the net-effect level, rather than component-level, the relationship between the measured BOLD signal and the evoked neural response.

A common approach for quantifying the neural-BOLD relationship is to manipulate a stimulus and measure the evoked electrophysiological and hemodynamic responses either simultaneously (*Brinker et al., 1999*; *Ngai et al., 1999*; *Logothetis et al., 2001*; *Devor et al., 2003*; *Sheth et al., 2004*; *Hoffmeyer et al., 2007*; *Huttunen et al., 2008*; *Magri et al., 2011*) or in separate experiments (*Hewson-Stoate et al., 2005*; *Nangini et al., 2008*; *Liu et al., 2010*). The hemodynamic signal measured can be either the fMRI BOLD signal itself or its physiological components (CBF, CBV, CMRO2). Several studies suggested that the BOLD signal or one of its components was linearly correlated with neuronal firing rates (*Smith et al., 2002*), synchronized synaptic activity as manifested in local field potentials (LFP) (*Ngai et al., 1999*; *Logothetis et al., 2001*; *Martindale et al., 2003*), electroencephalographic (EEG) (*Brinker et al., 1999*; *Arthurs et al., 2000*), magnetoencephalographic (MEG) (*Ou et al., 2009*), or intracranial electrocorticography (ECoG) (*Siero et al., 2013*) activities. However, some of these studies (*Ngai et al., 1999*; *Nangini et al., 2008*) may not have had sufficient statistical power for detecting departures from linearity. Other studies had a limited stimulation range, lacking measurements near the response threshold (*Logothetis et al., 2001*), or limited sampling points within a range (*Smith et al., 2002*). As a result, their findings of linearity may not be general. More importantly, a majority of the linear fits (*Ngai et al., 1999*; *Logothetis et al., 2001*; *Schumacher et al., 2011*) had a significant non-zero intersect, which would require a paradoxically non-zero change in the BOLD signal when the evoked neural response is zero. Absent of other factors, such a non-zero intersect implies that the underlying response function is actually nonlinear.

Other studies have suggested that the relationship between the BOLD signal and neural activity is nonlinear (*Devor et al., 2003*; *Sheth et al., 2004*; *Hewson-Stoate et al., 2005*; *Hoffmeyer et al., 2007*; *de Zwart et al., 2009*; *Liu et al., 2010*; *Magri et al., 2011*; *Kay et al., 2013*). *Magri et al. (2011)*, for example, found that the visually evoked fMRI BOLD responses as a function of band-limited neural signals (LFP and MUA) are sublinear. Others (*Devor et al., 2003*; *Sheth et al., 2004*; *Hewson-Stoate et al., 2005*) reported supra-linear relationships between neural activity and hemodynamic responses measured with intrinsic signal optical imaging and spectroscopy, or equivalently a linear relationship between the luminance contrast of a stimulus and the optical signal (*Lu and Roe, 2007*) (since the relationship between luminance contrast and neural activity is generally sublinear). This discrepancy could be due to the nonlinear relationships between the BOLD signal and its physiological components (CBF, CBV, and CMRO2) that were measured. Biophysical models relating the BOLD response to its constituents (*Buxton et al., 1998*; *Davis et al., 1998*; *Hoge et al., 1999*; *Griffeth and Buxton, 2011*), which are generally nonlinear, are needed to resolve this discrepancy.

Several studies (*Huettel and McCarthy, 2000*; *Liu et al., 2010*) used temporal summation to investigate the relationship between neural response and the BOLD signal. This is a popular non-invasive approach. The results, however, can be confounded with neural adaptation (*Huettel and McCarthy, 2000*). To avoid neural adaptation, the summation stimuli must be temporally separated with a sufficiently large lag time, thus precluding these methods from gauging the true (simultaneous) response nonlinearity.

## Our approach

Our approach is unique in that it allows a simple additive mixing of neural response across any time interval, including zero lag, to make a net-effect level quantification of the underlying relationship between the measured fMRI BOLD signal and the evoked neural response. Our method bypasses any nonlinearity between the stimulus and neural response. It also sidesteps the need to define the constituents of neural response (e.g., spiking activity, different frequency bands of LFP). These unique abilities stem from the configuration of the visual cortex in human achiasma: that there are two identical but independent populations of neurons that are co-localized on the cortex, sharing the same local control of blood supply. Each of these neuronal populations is associated with a distinct population receptive field. If stimulating one population results in some level of neural response, then for most reasonable definitions of 'neural response', stimulating both populations by placing identical stimuli in each of the two population receptive fields will double the local neural response (because two populations of neurons, instead of just one population, are responding). This notion of 'neural response' does not differentiate the specific components of neural response, such as spikes or LFP, which are themselves related in a complex manner. Rather, it is a holistic quantity (Z) that describe the totality of neural response, a notion that is perhaps more useful for most cognitive neuroscience investigations using fMRI. The fact that we succeeded in using the inferred BvZ functions to predict BOLD contrast-response functions from single-unit contrast-dependent spiking activity demonstrates the validity of our approach. While these results may suggest that Z is more related to spikes than to other neural response components nonlinearly related to spikes, more studies will be needed to quantify the link between the totality of neural response of a neural population and its individual components. The balance of the components that make up Z (e.g., spikes, LFP's) may depend on the stimulus, brain area, and task.

## Origins of co-locating but non-interacting neuronal populations

We found that each voxel in the low-level visual cortex of the achiasmic subject S has two non-interacting population receptive fields (pRFs). The empirical evidence we reported here suggests that there are two independent neuronal populations co-localized in each voxel of V1-V3 that subserve these two population receptive fields. *Williams et al. (1994)* found that in achiasmic Belgian sheepdogs, retinal axons originating from the nasal hemi-retina, which would normally cross the midline, instead innervated ipsilateral LGN and occupied those layers that would otherwise receive inputs from the contralateral eye. Alternating layers of the LGN in achiasma thus form mirror-image maps of the visual field. If axon guidance between LGN and V1 during development is typical in achiasma, then what would develop into the ocular dominance columns of V1 in a normal visual system will instead become visual-field dominance columns in achiasma, as suggested by *Victor et al. (2000)*. Preliminary measurements with 7T fMRI showed results consistent with the presence of visual-field dominance columns in V1 (*Olman et al., 2014*).

We expect the organization of visual-field dominance columns to be mostly within Layer 4 of V1, which receives direct hemifield-segregated inputs from LGN (*Williams et al., 1994*). Downstream from Layer 4, we expect this columnar organization to dissolve into finely intermingled but non-interacting unilateral (and monocular) neurons, for the following reason. In a normally developed visual system, binocular neurons receive inputs from monocular neurons in the neighboring ocular dominance columns. However, in achiasma, such 'binocular' neurons would stay monocular and unilateral. This is because the inputs from neighboring visual-field dominance columns would not correlate, as they represent unrelated locations in the left and right visual fields. Neurons that received inputs from neighboring columns would likely suppress or eliminate the inputs from one of the visual fields, chosen at random and without preference, since the inputs from the two visual fields are of equal strength (*Figure 1—figure supplement 1*, *Figure 3B*, *Figure 3—figure supplement 1*, *Figure 4—figure supplement 2*). Hence, downstream from Layer 4 of V1, neurons that represented different hemifields of the visual space would be functionally independent but finely intermingled.

In other words, we speculate a coarse segregation, at the scale of the ocular dominance columns, of the unilateral neuronal populations in Layer 4 of V1. The intermixing of the unilateral neurons would become finer downstream from V1 Layer 4, starting with neurons in the superficial and deep layers of V1. This may explain why the γ value of the BvZ function observed in V1 was higher than those found in V2 and V3 – the coarsely segregated neuronal populations in V1 Layer 4, which would

not fully share the local vascular system, lead to a more linear summation (higher value of γ). In V1, Layer 4 tends to have a larger contribution to the measured BOLD signal because of its dense vasculature but contributions from superficial and deep layers are also significant (*Polimeni et al., 2010*). We postulate that neurons representing the two hemifields would be more intermingled in the superficial and deep layers than those in Layer 4. These neurons would likely be more complex and less linear in their response properties to visual stimulation than neurons in Layer 4 (*Movshon et al., 1978a,b*). Their contribution to the BOLD signal relative to those in Layer 4 may be significantly reduced with long-duration simulation because of adaptation. The summation function thus appeared more linear in the 6-s condition than in the 1-s condition. It is important to point out that the summation function cannot be used to infer the BvZ function if the conditions of co-localization and non-interaction are not met. The condition of co-localization was not met in V1, and we could not use the summation function of V1 to infer its BvZ function. We do not think that the relationship between neural and BOLD responses depends significantly on stimulus duration. The fact that the BvZ functions inferred from V2 and V3, where the co-localization condition was met, provides an excellent fit to the V1 single-unit data (*Figure 5A*) further suggests that the inferred BvZ function is applicable in at least V1-V3.

## Conclusion

We have shown that the unique organization of the low-level visual cortex in human achiasma provides a versatile in vivo model for non-invasive quantification of the relationship between the evoked neural and fMRI BOLD responses. The presence of two independent neuronal populations with non-overlapping receptive fields at the same cortical location allows for the independent control of two separate sources of the local neural activity via stimulus presentation. By measuring the fMRI BOLD responses associated with a doubling of the local neural response, we found that the amplitude of the evoked BOLD response is proportional to the sum-total of the evoked neural response raised to a power around 0.5.

## Materials and methods

### Contrast detection

The achiasmic subject detected a Gabor target on a calibrated and gamma-linearized CRT display using only his right eye. The display had 10 bits of luminance resolution after gamma linearization by means of an analog video attenuator (*Li et al., 2003*) and custom software (https://github.com/usc-tlab/LinearFineContrast.git). In separate blocks of trials, the target was presented in the upper-left or lower-right quadrant against a uniform gray field with a luminance of 30.4 cd/m$^2$. A high contrast (Weber contrast = 1.0) task-irrelevant flickering checkerboard was displayed at a mirror-symmetric location from the target, either about the vertical meridian, such that the target and the flickering checkerboard would activate the same set of cortical regions, or about the horizontal meridian, such that the target and checkerboard would activate two different sets of cortical regions. The target was a 45° oriented Gabor with a center spatial frequency of 1 cycle/degree and a space constant ($\sqrt{2}\sigma$) of 1.4°. The task-irrelevant flickering checkerboard was of size 10° by 10° with 0.5° by 0.5° checks and flickered at 4 Hz in a square pattern. A two-interval-alternative-forced-choice paradigm (judging which interval contained the target), controlled with the adaptive procedure QUEST (*Watson and Pelli, 1983*), was used to measure the contrast threshold for a 75% correct detection rate. Each trial consisted of two temporal intervals (target-present and target-absent in random order) separated by a 500 ms blank. In each interval, the flickering checkerboard appeared for its entire duration of 600 ms. The Gabor target appeared for the last 150 ms of the target-present interval. During both intervals, a beep was presented 450 ms after the interval onset to reduce temporal uncertainty about target. The subject identified the target-present interval with a button press. A new trial began 500 ms after the subject had responded. Feedback was given after each response. For each target location, the subject completed 4 blocks of 50 trials.

## FMRI data acquisition and preprocessing

The fMRI data were collected using a 3-Tesla Siemens TIM Trio scanner with a 32-channel head coil using a T2*-weighted echo planar imaging sequence (TE/TR/flip angle = 25 ms/1 s/60°). 19 slices with 3x3x3 mm$^3$ isotropic voxels were prescribed to be perpendicular to the calcarine sulcus, covering the occipital pole. A high-resolution T1-weighted anatomical data set (3D MPRAGE; 1x1x1 mm$^3$ isotropic voxels, TE/TR/flip angle/TI = 2.98 ms/2300 ms/9°/900 ms) was collected in the same session before the functional runs and used to assist prescription of the functional slices. Eye movement was monitored online using an MRI compatible ASL 504 eye tracker with long-range optics. We analyzed the recording to estimate nystagmus amplitude and frequency during tasks and to determine if there were any stimulus-dependent biases in fixation. Additional offline measurements of eye movements were performed with a EyeLink 1000 Tower Mount monocular eye tracker.

Data analysis was performed using BrainVoyager QX (Brain Innovation, Maastricht, The Netherlands) and custom-built Matlab (MathWorks, Massachusetts, USA) code. The anatomical volume obtained in the retinotopic mapping session was segmented and inflated to generate a model of the cortical surface. Functional volumes were preprocessed, which included 3D motion correction, linear trend removal, and high-pass (> 0.0118 Hz) filtering. The functional images were aligned to one another and to the anatomical volume. The first 15–22 s of each scan (retinotopy experiment: 15 s, long-duration adaptation experiment: 22 s, BOLD summation experiments: 22 s for 6-s presentation and 17 s for 1-s presentation) were discarded to ensure equilibrium in longitudinal magnetization and subject state.

## Retinotopy

Rotating wedges and expanding half-rings made up of flickering (4 Hz) radially scaled color checkerboard patterns, presented to one eye at a time, were used to identify the retinotopic visual areas in subject S. For polar angle mapping, a 45° wedge with a radius of 8.5° rotated (jumped) counterclockwise by 11.25° every second, so that it swept the whole visual field in 32 s. For eccentricity mapping, the half rings were presented in the subject's right or left visual field in alternating blocks. They expanded in equal logarithmic steps from the center of display, where the subject fixated, and took 20 s to reach the maximum radius of 8.5°. The fixation of the display changed from '+' to '×' or vice versa randomly between 5 and 10 s (uniform distribution); the subject pressed a button as soon as a change of the fixation mark was detected.

## Defining regions of interest (ROIs) for the fMRI data analyses

A common set of regions of interest (ROIs) in V1-V3 was used to analyze data from both the adaptation and BOLD summation experiments. The stimuli used for the adaptation experiments (Gabor patches, see next subsection), set at 100% contrast, were also used to define these ROIs during scans independent of the main experiments. The stimuli consisted of four Gabor patches arranged in configurations resembling a forward or backward slash (*Figure 2C*). Each scan consisted of 13 fixation-only blocks interleaved with 12 stimulus blocks. The fixation-only blocks lasted for 16 s and the stimulus blocks lasted for 6 s. One stimulus configuration was presented in each stimulus block, and configurations alternated between blocks. During the stimulus block, the Gabor patches counter-flickered at 1 Hz and alternated their orientations (horizontal or vertical) at 0.5 Hz. The subject attended the fixation mark to perform a change-detection task at fixation.

A general linear model (GLM) assuming a stereotypical hemodynamic response function was used to identify the ROIs. The ROIs for backward- and forward-slash arrangements were defined separately, corresponding to the areas that responded more strongly to the ROI-defining stimulus than the blank interval (False Discovery Rate (FDR) < 0.05) and were within areas V1-V3 at the expected eccentricity (based on retinotopy scans). Since the two sets of ROIs (one for each stimulus arrangement) were very similar (*Figure 1—figure supplement 1*), we defined the stimulus response areas as the overlap of the two sets at FDR < 0.05. Because subject S has a nystagmus (horizontal amplitude generally less than ± 2.1°, peak-to-peak), we further restricted the ROIs to include, based on retinotopic coordinates, only the responsive voxels that responded to the central 2° of the two outer patches, which were 4° in diameter and centered at an eccentricity of 7°. The restricted stimulus response areas were further partitioned into ROIs in V1-V3 based on the subject's retinotopy.

## Long-duration fMRI adaptation experiment

The adaptor and test stimuli consisted of four luminance-contrast defined Gabor patches in two spatial configurations, resembling either a backward (upper-left to lower-right) slash or a forward (upper-right to lower-left) slash (*Figure 2C*). The two outer Gabor patches were at an eccentricity of 7°, with a carrier frequency of 1 cycle/degree and a space constant ($\sqrt{2}\sigma$) of 1.4°. The two inner Gabor patches were centered at an eccentricity of 3.5°, with a carrier frequency of 2 cycles/degree and a space constant of 0.7°. The Weber contrast of the carrier was 1.0 against a gray background of 156 cd/m$^2$. The carrier of all four Gabor patches could be oriented either horizontally or vertically.

Across different scans, the adaptor stimulus could be in one of four possible conditions: either in a forward- or backward-slash configuration, and oriented either horizontally or vertically. Relative to those of the adaptor stimulus, the four Gabor patches of the test stimulus were presented either at the same or different (mirrored) spatial location, with either the same or orthogonal orientation. There were 16 scans, with each scan consisting of one adaptor stimulus and 51 trials. A scan began with 20 s of pre-adaptation. Each trial started with a 5 s top-up adaptation period, followed by one of the five event types: four types of test stimulus (2 configurations by 2 orientations) or a blank interval. In a test trial, after 0.4 s of a blank interval following the top-up adaptation period, the test stimulus was presented for 0.3 s, followed by a 0.3-s blank. In a blank trial, no stimulus was presented for 1 s after top-up adaptation. The central fixation mark was present for all trial types (including the blank trial). At random intervals uniformly distributed between 3 and 7 s, the fixation mark randomly changed from '+' to '×' for about 400 ms before returning to '+'. The subject was instructed to attend to the fixation mark throughout a scan and press a key as soon as he detected a change of the fixation mark. The trial types were counterbalanced to ensure that the frequency of the immediately preceding trial type was equal, and the same for each of the 5 trial types.

For each experimental scan, the fMRI voxel data were % -transformed ($(y(t) - \overline{y})/\overline{y} \times 100\%$) and averaged within each ROI. ROI-averaged data were concatenated across scans, and the results were deconvolved against an indicator function formed by placing a Dirac delta function at the onset of the stimulus interval of the same trial type. Motion correction parameters were entered as regressors of no interest in the analysis to capture the noise introduced by head motion. The deconvolution analysis resulted in one response time course for each non-blank trial type, depicting the BOLD response evoked by the test stimulus.

## BOLD summation experiment

The BOLD summation experiment had three types of stimuli. Stimulus types A and B each consisted of 4 circular patches of black-and-white checkerboard arranged in either a backward-slash (A) or a forward-slash (B) configuration. For stimulus type A+B, the stimuli of type A and B were presented simultaneously, resulting in 8 checkerboard patches on the display. All patches in a stimulus were of the same contrast. We tested five Weber contrast levels from 0.05 to 1.0 in equal logarithmic steps against a uniform gray field of 156 cd/m$^2$. The outer patches were centered at an eccentricity of 7° and had radius 2°. The inner patches were centered at an eccentricity of 3.5° and had radius 1°. Each scan tested a specific contrast level. Within a scan, 19 stimulus blocks of 6 s each (or 1 s in a separate experiment) were interleaved with 19 16-s blank blocks. For each stimulus block, a stimulus of given type (A, B, or A+B) was presented, counter-flickering at 2 Hz. For each scan, the stimulus blocks were arranged in an otherwise random sequence with the constraint that each of the three stimulus types was preceded by every stimulus type equally often. To encourage and maintain fixation throughout a scan, the central fixation mark changed from '+' to '×' for about 400ms and back at random intervals, uniformly distributed between 3 s and 7 s. When the fixation was changed to '×' from '+', it was further tilted by ± 5°. The subject was to attend the fixation mark and report the direction of tilt as soon as the ' ×' appeared by pressing one of two response keys.

The time course of the evoked BOLD response for each stimulus condition was inferred using deconvolution. For the experiment with the 6-s stimulus duration, the deconvolution procedure was essentially identical to that used for the long-duration adaptation experiment.

For the experiment with the 1-s stimulus duration, the signal-to-noise ratio (SNR) of the data was low. To improve SNR, we used a GLM-based denoising method described in *Kay et al. (2013)*. The method assumes that physiological noise across measured voxels in the brain is correlated in the sense that it resides in a low-dimensional space. The method estimates this space in terms of its

principle components from the voxels that are not driven by the stimulus. These components are then entered into the GLM design matrix for a given experiment as regressors of no interest in order to estimate and reduce the influence of noise from each voxel. To ensure that this method does not introduce confounds, we further tested this method using data from the 6-s experiment, which are of high SNR, and found that results were essentially the same as the ones obtained with the generic deconvolution method (*Figure 3—figure supplement 2*).

## Model-neutral data stitching algorithm used to infer the BvZ function

Given *n* pairs of BOLD response amplitudes $B_{1,i}, B_{2,i}, i = 1 \ldots n$, corresponding to the *n* (*n*=5 contrast levels in our experiment) of single-sided and double-sided measurements, we sought to estimate *n*-1 parameters, $1 \leq Z_2 \leq \ldots \leq Z_n$, such that the 2*n* points $\{(1, B_{1,1}), (2, B_{2,1}), (Z_2, B_{1,2}), (2Z_2, B_{2,2}), \ldots, (Z_n, B_{1,n}), (2Z_n, B_{2,n})\}$ are maximally consistent, in the least squares sense, with a monotonic and smooth function. The multiplier 2 represents our assumption that the double-sided condition generated twice the neural response of the corresponding single-sided condition. In our case, the candidate monotonic and smooth function turned out to be a general cubic function restricted to be monotonically ascending. This is based on the following consideration. To be model-neutral, we first considered the entire family of cubic spline functions with *k* control points (*k*≥2) that are constrained to be monotonic. A cubic spline function with *k* control points has 2*k* degrees of freedom. We iteratively estimated the *n*-1 values of $Z_i$ and the 2*k* parameters of the cubic spline function by minimizing the normalized chi-square, which is the sum of squared errors between the 2*n* points and the cubic spline function, divided by the degrees of freedom of the residual. The degrees of freedom of the residual are $n + 1 - 2k$, and *k* has to be at least 2. In our case, *n*=5, meaning that the residual would have no degrees of freedom for *k*>2. As a result, we considered only the family of cubic spline functions with two control points, situated at $(1, B_{1,1})$ and $(2Z_5, B_{2,5})$, respectively.

The parameter estimation started with an arbitrary choice of $1 \leq Z_2, \ldots, \leq Z_n$, which was iteratively optimized by using the fminsearch function of MATLAB (The MathWorks). In the inner loop, we used the SLM MATLAB toolbox by John D'Errico (*D'Errico, 2009*) to fit a monotonic cubic spline function to the 2n points $\{(1, B_{1,1}), (2, B_{2,1}), (Z_2, B_{1,2}), (2Z_2, B_{2,2}), \ldots, (Z_n, B_{1,n}), (2Z_n, B_{2,n})\}$. We computed normalized chi-square, which we used as the objective function for fminsearch to choose the next set of $Z_i$ until convergence.

## Acknowledgements

We want to thank Stephen Engel for suggesting a comprehensive measurement of the contrast response function, Denis Pelli for the specific approach we used to quantify and visualize how our conclusion fits with data in the literature, David Heeger and Elisha Merriam for asking us to check for any effect due to anticipatory and other endogenous responses, Wilson Geisler for providing the analytical details used to generate Figure 2 in Heeger et al., (2000), Cheryl Olman, David Ress and Jonathan Winawer for probing discussions related to the BOLD signal, and Judith Hirsch, David Brown and Rachel Millin for their indispensable comments on the many drafts. We would also like to thank Jonathan Victor and two anonymous reviewers for their insightful and constructive comments as well as Jody Culham (Reviewing Editor) and Timothy Behrens (Senior editor) for their exceedingly helpful editorial instructions.

## Additional information

### Funding

| Funder | Grant reference number | Author |
| --- | --- | --- |
| National Eye Institute | NIH/NEI R01-EY017707 | Bosco S Tjan |
| National Science Foundation | NSF BCS-1255994 | Bosco S Tjan |

The funders had no role in study design, data collection and interpretation, or the decision to submit the work for publication.

## Author contributions

PB, Conception and design, Acquisition of data, Analysis and interpretation of data, Drafting or revising the article; CJP, Acquisition of data; Drafting or revising the article; BST, Conception and design, Analysis and interpretation of data, Drafting or revising the article

## Author ORCIDs

Bosco S Tjan, http://orcid.org/0000-0003-1290-5811

## Ethics

Human subjects: The Institutional Review Board of the University of Southern California approved the experimental protocol, and each subject provided written informed consent.

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
