## [Decision Letter]

Thank you for submitting your work entitled "Using an achiasmic human visual system to quantify the relationship between the fMRI BOLD signal and neural response" for peer review at *eLife*. Your submission has been favorably evaluated by Timothy Behrens (Senior editor) and three reviewers, one of whom is a member of our Board of Reviewing Editors (Jody Culham) and another is Jonathan Victor.

The reviewers have discussed the reviews with one another and the Reviewing editor has drafted this decision to help you prepare a revised submission.

Summary:

In a clever, elegant, and detailed set of experiments, this study uses a rare individual with achiasma (lack of an optic chiasm) to address how the fMRI (blood oxygenation level dependent or BOLD) signal summates. In achiasma, mirror-symmetric locations of the left and right visual fields are mapped to the same cortical location in visual areas. However, as careful psychophysics and fMRI adaptation experiments here show, stimulation at one site has negligible behavioral or neural effects on processing at the mirror-symmetric site. As such, achiasma provides a unique natural case study to investigate BOLD summation between effectively independent neural populations. The paper shows that doubling the stimulation sites in an achiasmic individual does not double the BOLD response; rather the BOLD response is approximately proportional to the square root of the neural response. These results are of general interest in neuroimaging considering the common use of BOLD fMRI to infer neural function.

Overall the referees were favorable, with all reviewers commenting favorably on the care and detail with which the project was executed.

Essential revisions:

Two main issues were raised by multiple reviewers and must be addressed in a revision.

1) First, two reviewers raised concerns about the imperfect mirror symmetry of the representations in the two visual fields. This must be addressed directly in the Discussion.

Reviewer 1/Reviewing editor:

The data in Figure 1 suggest that the representations across visual fields are largely but not perfectly homotopic (i.e., most points lie below a slope of 1 – and the data look similar across visual areas). Thus one concern is whether the choice of exactly homotopic locations may mean a slight displacement between the retinal locations stimulated, which could affect the data. If I were a stickler, I'd suggest an experiment in which stimulation was delivered to the visual location in the opposite field as determined by fMRI mapping). However, *eLife* policy is that if new data are essential to the conclusion, the paper will be rejected; in this case, I'm not sure the data is strictly essential. However, this concern needs to be noted and discussed.

Reviewer 3:

The assumption of colocalisation: Large-scale co-localisation of opposing hemifields is clearly demonstrated in the data. However, the representations of the relatively small mirror-symmetrical patches used might not fully overlap due to slight shifts in the representations. This is actually suggested by the eccentricity representations of both hemifields in Figure 1, where e1 exceeds e2 for by far most voxels in all visual areas examined. This potential problem needs to be addressed directly in the manuscript, e.g. a conservative ROI definition might help here. A related issue are potential partial voluming effects for the comparatively large voxel size used (3 mm isotropic), which might include cortical areas that are not driven by the stimulus, e.g. potentially negative BOLD effects.

2) Two reviewers raised concerns about discrepancies between visual areas. This must be addressed in the revision.

Reviewer 2:

My main substantive concern is the way that the authors present and interpret the discrepancy between V1 and the other areas, V2 and V3. Specifically, for V2 and V3, the confidence limits for the power law exponent include 0.5, and are similar for 6-sec and 1-sec presentations. But for V1, the 6-sec presentation yields a power law of about 0.75, with confidence limits that don't overlap with those for V2 and V3. The 1-sec V1 measurement is like V2 and V3, with confidence limits centered on 0.5. The authors interpret the finding that the 6-sec V1 measurement is closer to 1 as a result of less-complete overlap of the vasculature between the columns. While this may be true, it is unclear why the 1-sec measurement doesn't also show the same effect. So if the non-overlap issue is to be suggested as a plausible explanation for the V1 vs. V2/V3 discrepancy, something needs to be said about why only the 6-sec measurement shows its effect. Whatever the authors do on this point, I think that it is important to add that the explanation is at least somewhat conjectural, and one does need to also maintain the possibility that V1 is somehow different in terms of Z-to-B coupling. The paper's core contribution remains valid and important – the Z-to-B coupling is clear sublinear, in contrast to previous claims.

Reviewer 3:

Treatment of V1 data: V1 data and V2/3 data do not fully match. Therefore, the authors decide to concentrate on the V2/V3 findings. Given the large voxel size used I am not convinced that the respective discussion on V1 hemifield dominance columns fully resolves this issue. The partial exclusion of the V1 data appears critical, especially as we are already dealing with a single case study and the authors draw fairly general conclusions from this data set.

Recommended changes or considerations:

The following points/suggestions should also be considered in the revision, though there is some flexibility for the authors to deal with these as they see fit.

1) The biggest limitation of the study is the small sample size (n=1). The authors nicely addressed potential concerns in the cover letter to the journal. I suggest briefly including some of these arguments in the Discussion of the manuscript.

2) The Reviewing editor found the hypothetical wording of the last paragraph of the Introduction peculiar. While it fulfills the authors aim – to focus the paper on the BOLD question rather that achiasma per se – it would flow better to bring up achiasma first, explain the cortical retinotopic organization, and then explain why it's an ideal model for this particular question.

3) The following comment of Reviewer 2 should be considered: “Secondarily, I would not emphasize the specifically ‘square root’ nature of the relationship; in the absence of a mechanistic basis for such a relationship, it is a phenomenological model, and the exponent just happens to be in the neighborhood of 0.5 (for V2 and V3)”.

4) The authors should consider the following suggestion from Reviewer 2: “I think it is possible to infer the value of the scaling exponent by a simpler procedure, with fewer assumptions. I hope the authors will consider doing this, as it bypasses a complex step that I think is inessential.

But I don't think it's an either-or matter – the suggested simpler approach goes directly to the power-law approximation; the authors' approach uses a spline for the contrast-response as a stepping stone to the power-law fit. One can argue that bypassing the contrast response function is an advantage (because of simplicity and reduced assumptions) or a disadvantage (since it misses an opportunity for contact with known physiology). So these approaches are complementary, and optimally, I think that the authors should retain their original analysis but also add the proposed analysis, as follows:

The suggestion for the simpler analysis is as follows. The starting point is the set of values (z_i, 2z_i) at which they have measured the BOLD responses b_i,1=b(z_i) and b_i,2=b(2z_i). If one assumes a power law b=z^g, then the ratio of a pair of B measurements directly gives the scaling exponent g: b(2z_i)/b(z_i)=2^g. So one can plot these ratios, b(2z_i)/b(z_i), as a function of the index i, and see if these ratios are independent of i – or alternatively, if a trend emerges. If there's no trend, then the power law approximation is valid, the inferred value of the exponent g is the average of these ratios (optionally weighted by their reliabilities). If a trend emerges (which I doubt, given the good fits in the paper), then a power-law relationship cannot hold”.

[Editors' note: further revisions were requested prior to acceptance, as described below.]

Thank you for resubmitting your work entitled "Using an achiasmic human visual system to quantify the relationship between the fMRI BOLD signal and neural response" for further consideration at *eLife*. Your revised article has been favorably evaluated by Timothy Behrens (Senior editor) and Jody Culham (Reviewing editor).

The manuscript has addressed the concerns raised in the original review but there is one new paragraph that requires revisions to enhance the readability. The problematic paragraph is the last one of the Discussion beginning, "Given their progeny of being binocular neurons…". The first sentence did not make much sense (what are "progeny of being binocular neurons"?). It also did not set up the main goal of the paragraph, to address the discrepancies between areas. Overall, this paragraph should be rewritten/edited to make the arguments clearer.

Once this paragraph is cleaned up, the paper will be accepted without further ado.

---

## [Author Response]

*Essential revisions: Two main issues were raised by multiple reviewers and must be addressed in a revision. 1) First, two reviewers raised concerns about the imperfect mirror symmetry of the representations in the two visual fields. This must be addressed directly in the Discussion.* Reviewer 1/Reviewing editor:

*The data in Figure 1 suggest that the representations across visual fields are largely but not perfectly homotopic (i.e., most points lie below a slope of 1 – and the data look similar across visual areas). Thus one concern is whether the choice of exactly homotopic locations may mean a slight displacement between the retinal locations stimulated, which could affect the data. If I were a stickler, I'd suggest an experiment in which stimulation was delivered to the visual location in the opposite field as determined by fMRI mapping). However,* eLife *policy is that if new data are essential to the conclusion, the paper will be rejected; in this case, I'm not sure the data is strictly essential. However, this concern needs to be noted and discussed.*

Reviewer 3:

*The assumption of colocalisation: Large-scale co-localisation of opposing hemifields is clearly demonstrated in the data. However, the representations of the relatively small mirror-symmetrical patches used might not fully overlap due to slight shifts in the representations. This is actually suggested by the eccentricity representations of both hemifields in Figure 1, where e1 exceeds e2 for by far most voxels in all visual areas examined. This potential problem needs to be addressed directly in the manuscript, e.g. a conservative ROI definition might help here. A related issue are potential partial voluming effects for the comparatively large voxel size used (3 mm isotropic), which might include cortical areas that are not driven by the stimulus, e.g. potentially negative BOLD effects.*

The apparent lack of perfect homotopy is more likely caused by the subject's nystagmus having a saw-tooth waveform that jerks to the right (and drifts to the left), which may led to a leftward bias in terms of the time-averaged gaze position when trying to maintain fixation. This is consistent with the observed direction of bias in the eccentricity plot.

This apparent lack of perfect homotopy or gaze control has been factored into the experimental design and does not affect our main findings. Specifically, as long as the same voxels were stimulated equally by both versions of the single-sided stimuli in the summation experiment, the inferred BvZ functions will remain valid. To ensure that this condition was met, we defined the regions of interest (ROIs), using independent scans, to include only voxels that were activated by *both* versions of the one-sided ROI-mapping stimuli, at FDR<0.05. We further restricted the ROIs to include just the responses to the central 2° of the two outer (and larger) patches of 4° in diameter. The smaller patches near the fixation mark in the stimulus were not used in the summation analysis. They served to help maintain a stable fixation. Most of these design decisions were stated in the manuscript. In the revision, we have added a paragraph specifically addressing these issues associated with the lack of a perfect homotopy when we describe Figure 1 (please see: "The slight but systematic deviation from perfect homotopy in terms of eccentricity (Figure 1 top row) is likely due to his asymmetric nystagmus waveform (with a leftward slow phase) […] fMRI voxels used in the primary analysis were jointly activated by both stimuli."

We have added supplemental information to further demonstrate that the inferred BvZ functions were not affected the lack of perfect homotopy. We do so by: (1) showing that the voxels in the ROIs were equally activated by both versions of the one-sided stimuli during the summation experiment (Figure 4—figure supplement 2), and (2) when we reduced the ROIs to include only voxels that were highly activated by both versions of the one-sided stimuli during the summation experiment (not the independent ROI-defining scans), the resulting BvZ functions and gamma values were essentially the same (Figure 4—figure supplement 2).

*2) Two reviewers raised concerns about discrepancies between visual areas. This must be addressed in the revision.* Reviewer 2:

*My main substantive concern is the way that the authors present and interpret the discrepancy between V1 and the other areas, V2 and V3. Specifically, for V2 and V3, the confidence limits for the power law exponent include 0.5, and are similar for 6-sec and 1-sec presentations. But for V1, the 6-sec presentation yields a power law of about 0.75, with confidence limits that don't overlap with those for V2 and V3. The 1-sec V1 measurement is like V2 and V3, with confidence limits centered on 0.5. The authors interpret the finding that the 6-sec V1 measurement is closer to 1 as a result of less-complete overlap of the vasculature between the columns. While this may be true, it is unclear why the 1-sec measurement doesn't also show the same effect. So if the non-overlap issue is to be suggested as a plausible explanation for the V1 vs. V2/V3 discrepancy, something needs to be said about why only the 6-sec measurement shows its effect. Whatever the authors do on this point, I think that it is important to add that the explanation is at least somewhat conjectural, and one does need to also maintain the possibility that V1 is somehow different in terms of Z-to-B coupling. The paper's core contribution remains valid and important –the Z-to-B coupling is clear sublinear, in contrast to previous claims.*

Reviewer 3:

Treatment of V1 data: V1 data and V2/3 data do not fully match. Therefore, the authors decide to concentrate on the V2/V3 findings. Given the large voxel size used I am not convinced that the respective discussion on V1 hemifield dominance columns fully resolves this issue. The partial exclusion of the V1 data appears critical, especially as we are already dealing with a single case study and the authors draw fairly general conclusions from this data set.

The estimated gamma value from V1 is statistically different from those estimated from V2 and V3. It may not be very clear from Figure 4 that the 95% CI of the 1-s condition of V1 does not include 0.5 or the estimated mean values from V2 and V3. Furthermore, the 95% CIs for V2 and V3 do not include the estimated value of V1. (Note that we are dealing with 95% CIs, not standard errors. In a pairwise comparison with alpha=0.05, the requirement is for the CI of one estimated value not to include the estimated mean of the other value, and vice versa. It does not require the two CIs be non-overlapping.) We added a dashed line at gamma = 0.5 to make clear these conditions for a significant difference were met.

As the reviewers pointed out, it is clear that stimulus duration had an impact on the gamma value estimated from V1 (but not other areas). A likely cause of this is that the hemifield dominance columns in V1 are concentrated in Layer 4, because the thalamic inputs from the hemifields are segregated (Williams et al., 1994). The hemifield representations in the superficial and deep layers are likely intermingled for the same reason they intermingled in V2 and V3. Moreover, the effect of adaptation is probably less in Layer 4 than in the superficial and deep layers. This means that the observed BOLD signal in the 6-s condition (stronger adaptation) was probably dominated by signal from Layer 4, thus showing greater linearity than the 1-s condition. We have added text in the Discussion to describe this hypothesis (paragraph three, subsection “Origins of co-locating but non-interacting neuronal populations”).

Our recent 7T work showed a hemifield dominance organization in V1, and signal was likely attributable to the middle layer of the cortex – the putative Layer 4 (Olman et al., 2014). However, we do not wish to include this preliminary result in the current manuscript, as the work is still ongoing.

*Recommended changes or considerations: The following points/suggestions should also be considered in the revision, though there is some flexibility for the authors to deal with these as they see fit. 1) The biggest limitation of the study is the small sample size (n=1). The authors nicely addressed potential concerns in the cover letter to the journal. I suggest briefly including some of these arguments in the Discussion of the manuscript.*

We agree and added a new paragraph in the Discussion (paragraph two).

*2) The Reviewing editor found the hypothetical wording of the last paragraph of the Introduction peculiar. While it fulfills the authors aim – to focus the paper on the BOLD question rather that achiasma per se – it would flow better to bring up achiasma first, explain the cortical retinotopic organization, and then explain why it's an ideal model for this particular question.*

We have revised as suggested by moving and integrating the first two paragraphs of the Results to the revised Introduction. This eliminates the hypothetical phrase and makes the achiamsic model for the answering the BOLD question more concrete.

*3) The following comment of Reviewer 2 should be considered: “Secondarily, I would not emphasize the specifically ‘square root’ nature of the relationship; in the absence of a mechanistic basis for such a relationship, it is a phenomenological model, and the exponent just happens to be in the neighborhood of 0.5 (for V2 and V3)”.*

We agree. We have replaced "square root" by either the specific gamma value or the phrase "approximately 0.5", wherever appropriate.

*4) The authors should consider the following suggestion from Reviewer 2: “I think it is possible to infer the value of the scaling exponent by a simpler procedure, with fewer assumptions. I hope the authors will consider doing this, as it bypasses a complex step that I think is inessential.*

But I don't think it's an either-or matter – the suggested simpler approach goes directly to the power-law approximation; the authors' approach uses a spline for the contrast-response as a stepping stone to the power-law fit. One can argue that bypassing the contrast response function is an advantage (because of simplicity and reduced assumptions) or a disadvantage (since it misses an opportunity for contact with known physiology). So these approaches are complementary, and optimally, I think that the authors should retain their original analysis but also add the proposed analysis, as follows: The suggestion for the simpler analysis is as follows. The starting point is the set of values (z_i, 2z_i) at which they have measured the BOLD responses b_i,1=b(z_i) and b_i,2=b(2z_i). If one assumes a power law b=z^g, then the ratio of a pair of B measurements directly gives the scaling exponent g: b(2z_i)/b(z_i)=2^g. So one can plot these ratios, b(2z_i)/b(z_i), as a function of the index i, and see if these ratios are independent of i – or alternatively, if a trend emerges. If there's no trend, then the power law approximation is valid, the inferred value of the exponent g is the average of these ratios (optionally weighted by their reliabilities). If a trend emerges (which I doubt, given the good fits in the paper), then a power-law relationship cannot hold”.

We have added this analysis as a supplementary figure (Figure 4—figure supplement 1) and refer in the main text to this alternative, which is more straightforward but less general, yet perfectly applicable to our data. We have now pointed out in the main text that Figure 4 shows those doubling lines being parallel in log-log (i.e. no significant interaction between stimulus contrast and stimulus sided-ness), which implicates a power-law function. The alternative analysis is self-contained. We plotted B(2z_i)/B(z_i) as suggested in a supplemental figure to show the lack of trend (backed up by stats). Finally, we estimated gammas from the mean ratios (stated in the caption of Figure 4—figure supplement 1.).

[Editors' note: further revisions were requested prior to acceptance, as described below.]

*The manuscript has addressed the concerns raised in the original review but there is one new paragraph that requires revisions to enhance the readability. The problematic paragraph is the last one of the Discussion beginning, "Given their progeny of being binocular neurons…". The first sentence did not make much sense (what are "progeny of being binocular neurons"?). It also did not set up the main goal of the paragraph, to address the discrepancies between areas. Overall, this paragraph should be rewritten/edited to make the arguments clearer.*

We apologize for the oversight leading to nonsensical topic sentence. The paragraph in question should have been better integrated with the preceding paragraph. We have revised the entire section (“Origins of co-locating but non-interacting neuronal populations”) for readability and made clear that we are stating a hypothesis.